

# MLS measurements of stratospheric hydrogen cyanide during the 2015-16 El Niño event

Hugh C. Pumphrey[1], Norbert Glatthor[2], Peter F. Bernath[3,4], Christopher D. Boone[4], James Hannigan[5], Ivan Ortega[5], Nathaniel J. Livesey[6], and William G. Read[6]

[1]School of GeoSciences, The University of Edinburgh, Edinburgh, UK
[2]Karlsruher Institut für Technologie, Institut für Meteorologie und Klimaforschung, Karlsruhe, Germany
[3]Old Dominion University, USA
[4]University of Waterloo, Canada
[5]National Center for Atmospheric Research, Bolder, CO, USA
[6]NASA Jet Propulsion Laboratory, California Institute of Technology, Pasadena, CA, USA

*Correspondence to:* Hugh Pumphrey (Hugh.Pumphrey@ed.ac.uk)

**Abstract.** It is known from ground-based measurements made during the 1982-83 and 1997-98 El Niño events that atmospheric HCN tends to be higher than usual during such years. The Microwave Limb Sounder (MLS) on Aura has been measuring HCN mixing ratios since launch in 2004; the measurements are ongoing at the time of writing. The winter of 2015-16 has seen the largest El Niño event since 1997-98. We present MLS measurements of HCN in the lower stratosphere for the Aura mission
to date, comparing the 2015-16 El Niño period to the rest of the mission. HCN in 2015-16 is higher than at any other time during the mission, but ground based measurements suggest that it may have been even more elevated in 1997-98. As the MLS HCN data are essentially un-validated, we show them alongside data from the MIPAS and ACE-FTS instruments; the three instruments agree reasonably well in the tropical lower stratosphere. Global HCN emissions calculated from the GFED (V4.1) database are very much greater during large El Niño events and are greater in 1997-98 than in 2015-16, thereby showing good
qualitative agreement with the measurements. Correlation between ENSO indices, measured HCN and GFED HCN emissions is less clear away from the 2015-16 event. In particular, the 2009-10 winter had fairly strong El Niño conditions and fairly large GFED HCN emissions, but very little effect is observed in the MLS HCN.

## 1 Introduction

Hydrogen cyanide (HCN) is a minor constituent of the atmosphere which is produced almost entirely from biomass burning.
The earliest detailed study (Cicerone and Zellner, 1983) suggested that the main sink should be chemical loss, that this should be significant in the stratosphere, but small in the troposphere, and that the sources and sinks were small enough that the mixing ratio in the troposphere (which is approximately 0.2 ppbv) should show only small variations. Cicerone and Zellner (1983) identified a number of potential sources, mostly related to combustion. Subsequent studies (e.g. Li et al. (2000, 2003)) show that both sources and sinks are larger than Cicerone and Zellner (1983) believed, that the main source is biomass burning
and that the oceans form a significant sink in the troposphere. As a consequence, the tropospheric mixing ratio is more variable than was suggested by Cicerone and Zellner (1983).



The first daily, global satellite measurements of HCN to be reported (Pumphrey et al., 2006) were made by the Microwave Limb Sounder (MLS) (Waters et al., 2006) on NASA's Aura satellite (Schoeberl et al., 2006). The standard HCN product at that time contained large and obvious systematic errors, especially in the polar regions. The data described by Pumphrey et al. (2006) were weekly zonal means, produced by an off-line algorithm. Recently, version 4 of the MLS data was released; the

standard HCN product in this version is much improved, and is usable at low latitudes between 68 hPa and 1 hPa.

In this paper we present the version 4 HCN data to date, including the large El Niño event of 2015-16. We show that HCN concentrations entering the stratosphere are almost twice as large as usual during the 2015-16 El Niño and are enhanced to some extent during some of the smaller El Niño events earlier in the Aura mission.

## 2   ENSO and HCN

In this section we describe the El Niño / Southern Oscillation system briefly and review the published measurements of HCN during previous large ENSO events.

### 2.1   The El Niño / Southern Oscillation

The El Niño / Southern Oscillation (ENSO) is a large-scale dynamical feature of the climate system and is the largest cause of inter-annual climate variability. A good introduction to the subject is provided by Sarachik and Cane (2010). The main

feature of ENSO is a non-periodic oscillation on a timescale of 2-7 years between two distinct phases. The El Niño phase is characterised by warmer water in the eastern Pacific and the movement and spread of a large region of deep convection from the western Pacific into the central Pacific. The La Niña phase is characterised by cooler water in the eastern Pacific and the confinement of the region of deep convection to the western Pacific. Although ENSO has its most obvious effects in the tropical Pacific, its effects are felt further afield. In particular, the El Niño phase is associated with droughts in Australia and Indonesia,

while the La Niña phase is associated with greater than normal rainfall in those areas. Larger El Ninõ events tend to begin in the middle of a calendar year and persist until well into the following calendar year.

It is common practice to summarise the state of ENSO at a given time by a single number called an ENSO index. There are a range of these indices in use. Some are determined purely from sea-surface temperature data; the Nino3.4 index (Barnston et al., 1997) is such an index and is taken from an area of ocean chosen specifically to make it useful as a general indicator of ENSO

state. Other indices are determined from purely atmospheric data; an example is the Southern Oscillation Index (SOI), defined as the difference between the normalised pressure anomalies at Darwin and Tahiti (Trenberth, 1976; Horel and Wallace, 1981). Yet other ENSO indices amalgamate several sorts of data into a single index; the Multivariate ENSO Index (MEI) (Wolter and Timlin, 1993, 1998) is a much-used example of this type. ENSO indices are usually calculated as the difference from the mean value over a base period, divided by the standard deviation of that same period. This ensures that they have a value of zero

when the climate system is not in either a El Niño phase or a La Niña phase and that the value lies in the range $\pm 1$ for much of the time, moving only occasionally outside the range $\pm 2$. Figure 1 shows time series of the MEI, SOI and Nino3.4 indices





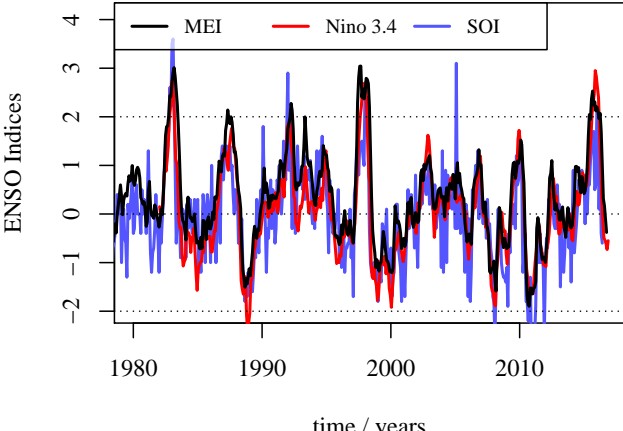

**Figure 1.** Time series of three different ENSO indices between 1977 and the present. The SOI as usually defined is negative during a positive ENSO phase; we have plotted the negative of the SOI for easier comparison with the other indices. SOI and Nino3.4 data obtained from http://www.cpc.noaa.gov/data/indices/. MEI data obtained from http://www.esrl.noaa.gov/psd/enso/mei

between 1977 and the present. It is clear that there is a high level of correlation between the indices. For the rest of this paper we will therefore use the MEI to indicate the phase of the ENSO system.

There have been three particularly large El Niño events between 1950 and now: in 1982–3, 1997–8 and 2015–16. All three are shown in figure 1 and are the only times several of the indices exceed 2 for several months. Only the last of these large El Niño events occurred during the Aura mission. There were, however, several smaller El Niño events during mission and the preceding two years, most notably in 2002–3, 2004–5, 2006–7, 2009–10 and 2014–15, the latter arguably being the initial part of the 2015-16 event.

It is known that El Niño events cause droughts and hence an excess of biomass burning in Indonesia and neighbouring countries. Field et al. (2016) report that the 2015-16 event was the most severe in the period when NASA's Earth Observing System (EOS) satellites were operating, making it significantly more severe than 2006-7. The 1997-8 event occurred before the EOS satellites began operating, but other types of observation suggest that it was even more severe than 2015-16.

## 2.2 HCN during ENSO events

Although there were no regular satellite measurements of HCN during the major El Niño events of 1982–3 and 1997–8, there were a number of ground-based measurements made using Fourier-transform Infrared (FTIR) spectroscopy. These measurements were made at the Jungfraujoch in Switzerland (46.5°N), (Rinsland et al., 1999), Mauna Loa in Hawaii (19.3°N), (Rinsland et al., 2000) and Kitt Peak, Arizona (32°N), (Rinsland et al., 2001).

The measurements from all three sites are shown in figure 2. (Data were obtained from the website of the Network for the Detection of Atmospheric Composition Change (NDACC) — http://www.ndsc.ncep.noaa.gov/.) Note that the data are the average mixing ratio for a partial column; the partial columns are for slightly different altitude ranges for the three measurement





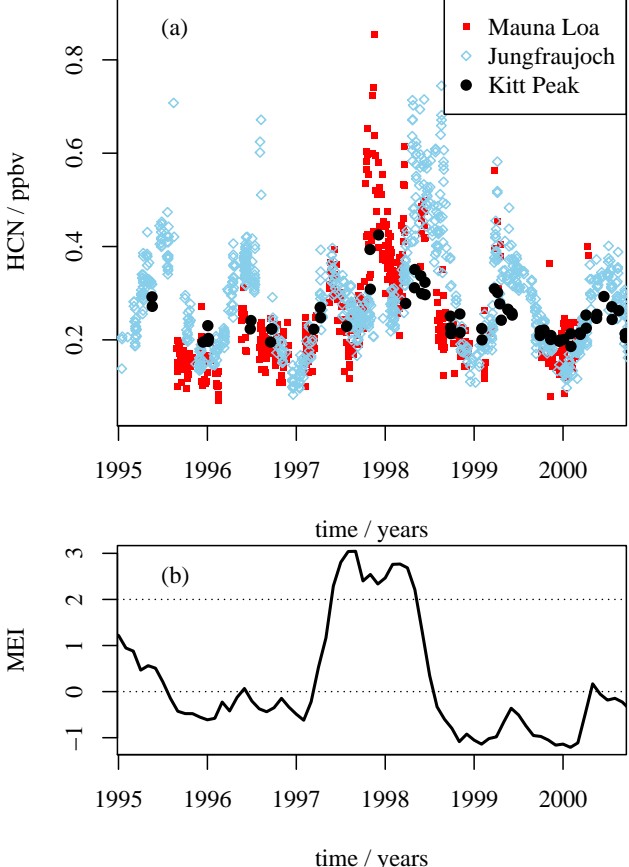

**Figure 2.** Ground-based HCN measurements between 1995 and 2000, at Kitt Peak (32°N), Mauna Loa (19.3°N) and Jungfraujoch (46.5°N), with the Multivariate ENSO Index (MEI) shown for comparison. Note that the values are slightly different from those shown in Rinsland et al. (1999, 2000, 2001) as the data have been re-processed with updated spectroscopy.

sites: 3.40–16 km for Mauna Loa, 3.58–11 km for Jungfraujoch and 2.09–14 km for Kitt Peak. All three sites show a substantial increase (at least a doubling) of HCN mixing ratios during the 1997–8 El Niño. The response is faster at the lower-latitude sites; the enhanced HCN is observed at Kitt Peak and Mauna Loa in late 1997 and does not appear at the Jungfraujoch until spring 1998. It is shown by Rinsland et al. (1999), using trajectory calculations, that the air arriving at Hawaii during the period

5 with enhanced HCN does so from the West; it is suggested that the HCN comes from the extensive forest fires which burned in Indonesia during late 1997, and that a similar explanation is plausible for the Kitt Peak data. The Jungfraujoch data are less easy to explain; Rinsland et al. (2000) suggest a variety of possible causes.





## 3 The MLS HCN data

### 3.1 The measurement and estimation procedure

The Aura satellite (Schoeberl et al., 2006) was launched in July 2004, and the MLS instrument (Waters et al., 2006) has operated with little interruption from August 2004 to date. The satellite orbits at an altitude of 705 km, performing approximately 14.5

orbits per day. The MLS instrument consists of a 1.6 m parabolic dish antenna feeding heterodyne radiometers operating at 118, 190, 240 and 640 GHz. A separate small antenna feeds another radiometer operating at 2.5 THz. The output of the radiometers is analysed by banks of filters. The antenna looks forward from the Aura platform, in the plane of the orbit, and is scanned across the Earth's limb 240 times per orbit. As the orbit is inclined at 98° to the equator, the instrument observes a latitude range from 82°S to 82°N every day. The observations are of thermal emission from the atmosphere and can therefore be made

day and night. The orbit is sun-synchronous, so the observations are always made at the same two local times for a given latitude; the ascending equator crossing time is 13:45. The radiances reported by the filter banks are used as input to a software package (Livesey et al., 2006) which estimates profiles of temperature, and of the mixing ratios of the target chemical species. Most MLS estimated profiles, including HCN, are reported on pressure levels spaced at 6 levels per pressure decade, a spacing of approximately 2.7 km in altitude. The estimated profiles are spaced 1.5° (167 km) apart along the orbit track. All MLS data

products are documented in some detail in Livesey et al. (2017).

Hydrogen cyanide is a linear molecule with a large dipole moment. Linear molecules have a simple microwave spectrum, with lines at integer multiples of a fundamental frequency: 88.6 GHz in the case of HCN. MLS has two filter banks (Band 6F and Band 27M) which are fed from the 190 GHz radiometer and are affected by the second of these lines, which lies at 177.26 GHz. Details of the radiances in this part of the spectrum and of the spectral coverage of the instrument, are given in

Pumphrey et al. (2006).

In previous data versions (version 1, 2 and 3) the MLS HCN data were retrieved as part of the same inversion as all other species retrieved from the 190 GHz measurements. In version 4 of the MLS data, HCN is estimated as part of a separate retrieval phase which uses only bands 6F and 27M as input. HCN mixing ratios are estimated over a range 0.1 hPa to 100 hPa. Initial inspection of the version 4 HCN suggests that it is similar to earlier versions at 32 km (10 hPa) and above, and is much

improved in the lower stratosphere. The main improvement is that the version 4 HCN does not have regions of persistently negative mixing ratio in the extratropical lower stratosphere. As a result of this, the data are recommended for use in the range 0.1 hPa to 21 hPa. In this paper we make cautious use of the data between 31 hPa and 100 hPa, justifying their use by comparison with other data where that is possible. The vertical resolution of the HCN data is considerably poorer than the 2.7 km spacing between the levels on which the data are reported. Averaging kernels for the retrieval are shown by Livesey

et al. (2017); the vertical width is between 7 km and 9 km in the lower stratosphere. Furthermore, the averaging kernel for 100 hPa has a poor shape, peaking at 68 hPa and having a far lower response at 100 hPa; this implies that the 100 hPa data reflects changes at higher altitudes. A full validation paper describing the data is planned. In the absence of a full validation of the data we show them alongside data from the MIPAS instrument on Envisat and the ACE-FTS (Atmospheric Chemistry Experiment - Fourier Transform Spectrometer) on SCISAT (Bernath et al., 2005; Bernath, 2017). HCN data from MIPAS have





been described recently (Glatthor et al., 2015), and are of good quality between 10 km and 30 km altitude, but are not available for dates after March 2012. We use MIPAS data versions V5H_HCN_21 (2002-2004), V5R_HCN_222 (2005-April 2011) and V5R_HCN_223 (April 2011 - 2012). A small amount of version V5R_HCN_120 is used to fill gaps in the later versions in 2005 and 2006. ACE-FTS data (version 3.5/3.6) (Boone et al., 2005; C. D. Boone and Bernath, 2013) are available over the

entire Aura mission. As ACE-FTS is a solar occultation instrument, measurements are made at only two latitudes on any one day; these latitudes change slowly from day to day, crossing the equatorial region approximately once per month (Bernath, 2017). The ACE-FTS HCN data during the 2015-16 El Niño event have been reported by Sheese et al. (2017).

## 3.2   The "tape recorder" signal

It was reported by Pumphrey et al. (2008), and confirmed by Glatthor et al. (2015), that HCN shows a tape recorder signal in

the tropical lower stratosphere. The tape recorder phenomenon was originally reported in water vapour measurements (Mote et al., 1996) and will be observable for any species which is chemically stable in the lower stratosphere and which has seasonal or interannual variations in the mixing ratio entering the tropical lower stratosphere. Attempts to model the tape recorder signal in HCN are described by Park et al. (2013), Pommrich et al. (2010) and Li et al. (2009). These studies show that the interannual variability in lower stratospheric HCN is driven by variability in biomass burning, and that much of that variability comes from

Indonesia and the surrounding region.

Figure 3 shows the tape recorder signal in the MLS, MIPAS and ACE-FTS data. The 2004–2008 period shows a strong biennial cycle, with particularly high values entering the stratosphere in late 2007 and most of 2008, as reported by Pumphrey et al. (2008). This feature is corroborated by the MIPAS and ACE-FTS data; in the MIPAS data it appears to extend back to 2002. The 2009–2013 period is rather featureless in the MLS data. The MIPAS data show more structure, with high values

at 100 hPa and 68 hPa at the end of 2009, 2010 and 2011. This implies that MLS has much poorer sensitivity than MIPAS at these altitudes. The ACE-FTS data in the 2009–2013 period shows an annual tape recorder signal; this is almost invisible in the MLS data due to the instrument's poorer vertical resolution. MIPAS data are unavailable after spring 2012, but the MLS and ACE-FTS data show more variability in the most recent part of the time series. There are particularly low entry values in early 2014, and the values entering the stratosphere in late 2015 and early 2016 are larger than any seen so far during the Aura

and Envisat missions. The onset of high values in late 2015 is very rapid and appears in the MLS data to be simultaneous at the 100 hPa, 68 hPa and 46 hPa levels. The ACE-FTS data are too sparse in time to show how rapid the onset is, but suggest that it initially affects only the 100 and 68 hPa levels. The rapid onset at 46 hPa in the MLS data is, again, due to the instrument's limited vertical resolution. Although the onset of high HCN values is rapid, it is not instantaneous, occurring over about 18 days, between 2015-10-20 and 2015-11-7. The smaller increase in late 2006 is similar, but occurs over a longer period:

approximately 45 days between 2016-10-18 and 2006-12-3.

## 3.3   HCN as a function of latitude

Figure 4 shows the MLS HCN data as a function of time and latitude. The El Niño years of 2015-16 and 2006-7 show clear increases in HCN. At 68 hPa, the sudden onset of the increase is seen towards the end of the first year of the event, approximately





**Figure 3.** Zonal mean HCN anomaly (in ppbv) for MLS, MIPAS and ACE-FTS, for the 12.5°S to 12.5°N latitude band, as a function of altitude and time. The MLS and MIPAS data shown are 20-day averages in order to reduce the noisiness of the figure. The ACE-FTS data are calendar month averages. For consistency, the anomalies are calculated relative to the 2006-2011 period, for all instruments. The very low values in the MIPAS data in 2002-4 above 25 km are thought to be at least partly an instrumental artefact.







**Figure 4.** MLS HCN, in ppbv, as a function of time and latitude, for three pressure levels. Note that the colour scale is different for each level.





$10°$ south of the equator. The subsequent maximum values occur in the middle of the second year of the event, close to the equator. At 46 hPa, the initial increase is less clear and much of the equatorial signal occurs some months later than at 68 hPa, as would be expected from figure 3.

The 100 hPa data shown in figure 4 appear very similar to the 68 hPa data. As noted above, it is likely that the 100 hPa data contain little useful information. To confirm this, we show in figure 5 a shortened version of the time series, with the MIPAS data for comparison. It is clear from this that the two instruments agree rather well at 31.6 hPa, but that the agreement becomes worse with decreasing altitude. The MIPAS data at 100 hPa (and to some extent at 68 hPa) show some notable features not present in the MLS data. In particular, the HCN entering the stratosphere during the early part of an El Niño event does so in two disconnected regions at about $30°$S and $30°$N. The southern region is just visible in the MLS data at the end of 2006, but is far clearer in the MIPAS data. Enhanced HCN seen at $30°$N during the northern hemisphere summer is caused by the flow of polluted air into the stratosphere via the Asian monsoon (Randel et al., 2010; Ploeger et al., 2017). This feature is difficult to discern in the MLS data. It is clear from the MIPAS data that the feature does not extend to any greater altitude than 68 hPa, implying that the MLS data contain little if any information from altitudes lower than 68 hPa.

The small area of high HCN values seen at $12°$S in February 2009, in figures 4 and 5, is caused by the Black Saturday fires in Australia (Pumphrey et al., 2011).

### 3.4 Location of the rapid increase in HCN

To further examine the rapid increase in HCN in October 2015, we show in figure 6 eight-day mean maps of HCN at 68 hPa. It is clear from these that the HCN arrives at this altitude close to the equator and at a very limited range of longitudes, close to $100°$E. The subsequent increase to a maximum value in mid-2016 (not shown) does not occur at an identifiable longitude; the distribution of HCN in December 2016 and after appears zonally symmetric. The smaller sudden increase in October and November of 2006 also occurs around $100°$E.

### 4 Discussion

The most detailed and comprehensive estimate of biomass burning and its effect on the atmosphere is the Global Fire Emissions Database, Version 4.1 (GFED4s), currently obtainable from http://www.globalfiredata.org. This is an updated version of the database described in van der Werf et al. (2010). The emissions data consist of the mass of dry matter $m_{DM}$ burned every month, in every $0.25°$ grid box. Each grid box is provided with metadata showing the fraction, $f$, of the box covered with each of six surface types (savannah, boreal forest, peat etc.). Also provided is a table giving an emission factor, $E$, for each combination of surface type and emitted molecule; we show an excerpt from this in table 1. With this information it is straightforward to





**Figure 5.** MLS (left) and MIPAS (right) HCN, in ppbv, as a function of time and latitude, for four pressure levels. Note that the colour scale is different for each level. A constant offset has been added to the MIPAS data at each level to remove the bias between the two instruments at the equator; the offsets are +0.04 ppbv at 100 hPa, -0.025 ppbv at 68.1 hPa, -0.032 ppbv at 46.4 hPa and -0.038 ppbv at 31 hPa,




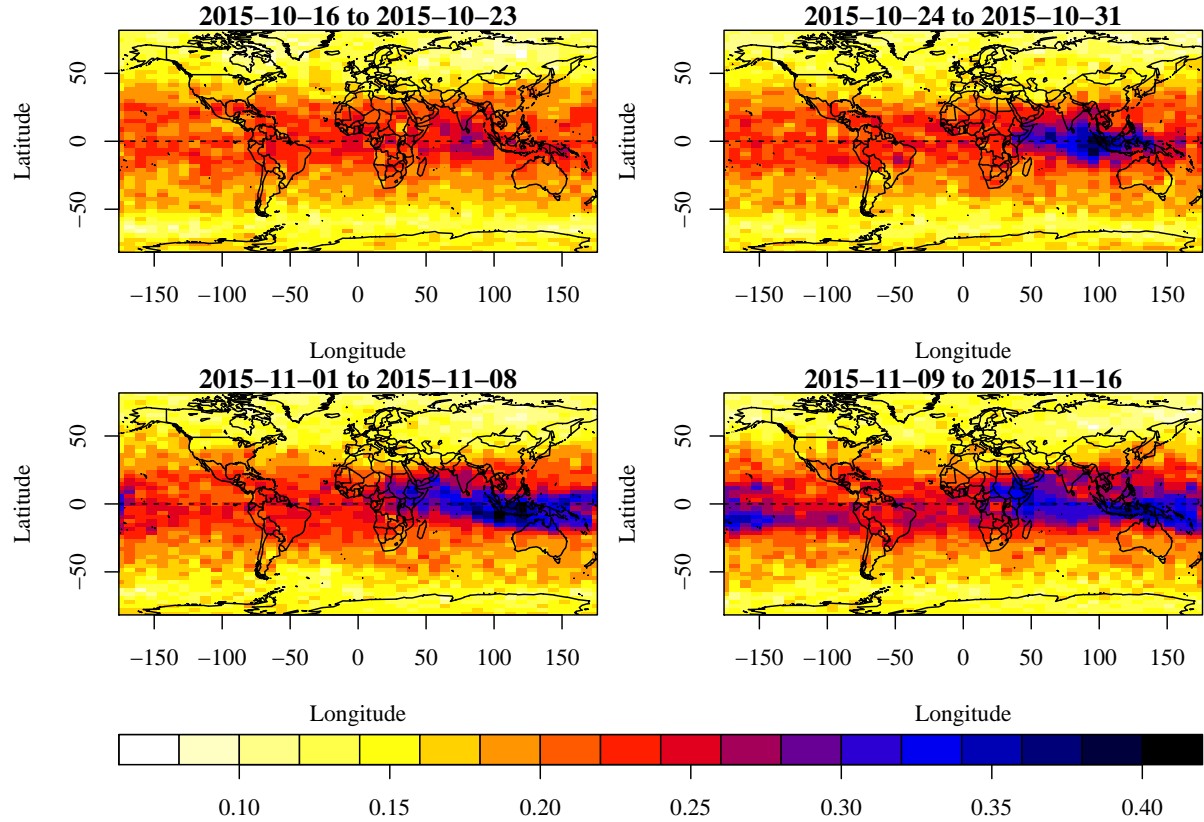

**Figure 6.** Eight-day average maps of MLS HCN, in ppbv, at 68 hPa.

**Table 1.** Emission factors (in $g\,kg^{-1}$) for use with GFED4s, for all surface types and for a selection of molecular species, including dry matter (DM). Note that C is total carbon.

|        | SAVA  | BORF | TEMF  | DEFO  | PEAT  | AGRI  |
|--------|-------|------|-------|-------|-------|-------|
| DM     | 1000  | 1000 | 1000  | 1000  | 1000  | 1000  |
| C      | 488.2 | 465  | 489.4 | 491.6 | 570.1 | 480.3 |
| $CO_2$ | 1686  | 1489 | 1647  | 1643  | 1703  | 1585  |
| CO     | 63    | 127  | 88    | 93    | 210   | 102   |
| HCN    | 0.41  | 1.52 | 0.72  | 0.42  | 8.11  | 0.29  |





calculate an estimate of the mass $m$, of any molecule emitted from any grid box in any month, by summing over the six surface types:

$$m_{\mathrm{HCN}} = m_{\mathrm{DM}} \sum_{j=1}^{6} f_j E_{j,\mathrm{HCN}} \tag{1}$$

Emissions from individual grid cells may be summed to give the emissions for any region of the world. For convenience,
GFED4s is provided with a mask dividing the world into 14 regions.

We show in figure 7 the monthly emissions of HCN and total C calculated for the whole world, the region between 30°S and 30°N, and for the Equatorial Asia (EQAS) region as defined by GFED4s; this region essentially covers the countries of Indonesia, Malaysia and Papua New Guinea. It is clear that the EQAS region accounts for a much larger fraction of HCN emissions than for biomass-burning emissions as a whole. This is because the EQAS region has a far larger fraction of the peat
surface type than other equatorial regions and because $E_{\mathrm{HCN}}$ is far larger for peat than for the other surface types.

In the strong El Niño year 2015-16 and the moderate El Niño years 2002-3, 2006-7 and 2009-10 there is a strong peak in global HCN emissions toward the end of the first calendar year; these emission peaks are almost entirely due to emissions from the EQAS region. In the La Niña years 2007-8, 2008-9, and 2010-11, global HCN emissions are lower, and are mostly due to parts of the world other than the EQAS region. The remainder of the time, when the MEI is close to zero, the EQAS region
makes a significant, but not dominant, contribution to global HCN emissions. This contribution often occurs in the NH autumn, as in El Niño years, but may occur at other times of year (e.g. in 2013 and 2005).

The HCN values in the lower stratosphere are, for the most part, what would be expected from the emissions. The emission spikes in autumn 2002, 2006, 2015 and 2014 all result in a sharp increase in HCN at 68 hPa, with a continued rise in the following months. In 2006, 2014 and 2015 the initial increase occurs over the EQAS region. We show this in figure 6 for 2015;
2006 and 2014 are similar but less spectacular. We can not extend this observation to 2002 as this was before the Aura launch, and the MIPAS data have a gap at the time of the sharp increase.

The anomalous year is 2009; this year shows a large peak in autumn HCN emissions from the EQAS region, but very little effect is seen in the stratosphere.

These observations support the hypothesis that the excess HCN in the stratosphere during an ENSO event is almost entirely
due to unusual amounts of biomass burning in the EQAS region. Rather more tentatively, we suggest that in some El Niño years (2006-7, 2014-15, 2015-16), a large proportion of the HCN is transported very rapidly to the stratosphere above the EQAS region; the remaining fraction being dispersed within the troposphere. In other El Niño years (2009-10) the rapid transport does not occur, and any HCN which reaches the stratosphere does so by a less direct route.

Was the 2015-16 El Niño as large an event, in terms of HCN, as the 1997-8 event? Figure 8 shows the GFED HCN emissions
and MEI; it is clear that, for HCN emissions in the GFED data, 1997-8 was more than twice as large as 2015-16. It is difficult to compare atmospheric HCN for the two events as the data available is so different. Fourier transform infrared spectrometers (FTIR) (see Fig. 2) provide the longest record, but few are in the tropics, and, of these, only the Mauna Loa station was recording during both the 1997-8 and the 2015-16 events. Figure 8 shows data from this instrument. FTIR instruments have



little vertical resolution, a profile usually having no better than two independent values. We therefore plot average mixing ratios over two deep layers. The FTIR data are hard to interpret by eye as they show a great deal of scatter. We therefore smooth them using the local linear interpolator (Wand and Jones, 1995), with a bandwidth of 0.05 years where the data are dense, and 0.45 years where there are large gaps. In the stratosphere, HCN in the 1997-8 event appears to start from a lower value, increase
to a higher value and stay elevated for longer, despite the fact that the peak values in 2015-16 is larger. In the troposphere, the 1997-8 appears larger, both in terms of the peak value and the smoothed curve. In neither the troposphere nor the stratosphere is the difference between 1997-8 and 2015-16 as large as one would expect from the GFED emissions estimates.

## 5   Conclusions

The main driver of interannual variability of HCN in the tropical lower stratosphere is variability in the only significant source:
biomass burning. In El Niño years, the equatorial Asia region tends to suffer severe draughts, leading to unusual amounts of biomass burning. In some years a significant fraction of the biomass burning products from these events are transported rapidly to the stratosphere, arriving at 68 hPa over the EQAS region. The 2015-16 El Niño event is the strongest event of this type during the Aura mission, resulting in mixing ratios of HCN in the lower stratosphere which are 45% higher than normal. Both the GFED emissions estimates and the limited data available from a ground-based FTIR instrument suggest that HCN
emissions were even greater in the 1997-8 El Niño event.

*Author contributions.*   HCP wrote most of the paper. NG advised on the MIPAS data and made a number of corrections to the text. PFB and CB advised on the ACE-FTS data. JH and IO advised on the use of the FTIR data, and expedited the availability of the Mauna Loa data from 2016. WGR advised as to how the HCN data are handled in the MLS data processing. NJL made a number of corrections to the text and suggested various improvements to the figures.

*Acknowledgements.*   The FTIR data used in this publication were obtained as part of the Network for the Detection of Atmospheric Composition Change (NDACC) and are publicly available (see http://www.ndacc.org). The ACE mission is supported primarily by the Canadian Space Agency. Work at the Jet Propulsion Laboratory, California Institute of Technology, was performed under contract with NASA. Work on MLS at Edinburgh has been funded by NERC under grant NE/E003990/1 and previous grants. The article processing charges for this open-access publication were paid by the RCUK Open Access Publication Fund.





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







**Figure 7.** (a) Zonal-mean HCN from MLS, MIPAS, and ACE-FTS at 68 hPa (about 18.5 km), between 12.5°N and 12.5°S. (b) Multivariate ENSO Index (MEI) for comparison. (c) GFED4s HCN emissions. (d) GFED4s C emissions. The equatorial region as shown in (c) and (d) is defined as the region between 30°S and 30°N.



**Figure 8.** (a) and (b): HCN data from the Mauna Loa FTIR instrument, for two different height ranges. (c) and (d): MEI and GFED HCN emissions for comparison.