# Peer review of "MLS measurements of stratospheric hydrogen cyanide during the 2015-16 El Niño event"

_Atmospheric Chemistry and Physics, 2017_

## Author Comment (AC1) · 13 Jul 2017

Owing to an oversight, the correct text regarding the FTIR instrument did not get inserted into the final copy submitted to the discussion stage. The text between page 12, line 30 and page 13, line 7 should have read as follows:
* * *
It is difficult to compare atmospheric HCN for the two events as the data available is so different. Fourier transform infrared spectrometers (FTIR) (see Fig. 2) provide the longest record, but few are in the tropics, and, of these, only the NDACC Mauna Loa station was recording during both the 1997-8 and the 2015-16 events (Hannigan et al., 2009). Figure 8 shows data from this instrument. Solar viewing FTIR instruments

derive vertical information from the pressure broadening of absorption transitions in transmission spectra and typical degrees of freedom in the HCN retrieval are between 2.5 and 3.5. Here we plot average mixing ratios over two deep layers. To reveal the observation of the El Nino enhancements and diminish short term variability we smooth the individual observations using the local linear interpolator (Wand and Jones 1995), with a bandwidth of 0.05 years where the data are dense, and 0.45 years where there are lapses in observations; these lapses occur intermittently between 1999 and 2012. Comparing the two events in the stratosphere, HCN in 1997-8 starts from a slightly lower background level near 0.2 ppbv, increases to a higher value above 0.3 ppbv and persists at an elevated level for longer. Despite peak values in 2015-16 being higher, higher sustained levels above 0.3 in 1997-8 appear to show greater stratospheric loading. In the troposphere, the 1997-8 event appears larger, both in terms of the peak value and the smoothed curve, in general agreement with GFED. In neither the troposphere nor the stratosphere is the difference between 1997-8 and 2015-16 as large as one might expect from the GFED emission estimates alone.

————————————————————————

The corrected text contains one extra reference.

Hannigan, J. W., Coffey, M. T., and Goldman, A.: Semiautonomous FTS Observation System for Remote Sensing of Stratospheric and Tropospheric Gases, J. Atmos. Oceanic Technol., 26, 1814– 1828, doi:10.1175/2009JTECHA1230.1, 2009.

————————————————————

---

## Referee Comment (RC1) · Anonymous Referee #1 · 28 Jul 2017

This paper "MLS measurements of stratospheric hydrogen cyanide during the 2015-16 El Niño event" by Pumphery et al. shows elevated levels of HCN in the lower stratosphere over the Equatorial Asia (EQAS) region from ground based measurements and satellite observations during 2015-16 El Niño event. The topic of the study is interesting and suitable for the journal. I suggest major revision before its publication.

General comments: Manuscript is very short and lack details

It is known that droughts related to El Niño events are associated with sinking branch of the Walker circulation. Therefore zonal mean plots in Figure-3 (averaged over 12.5N-12.5S) provide poor information about vertical propagation of HCN in the stratosphere. During El – Niño event injection into the stratosphere may be along rising branch of the Walker circulation. This feature gets averaged in the zonal plots. A major concern

is that the authors state that source of HCN in the lower stratosphere is from Maritime Continent. While during EL-Nino there is subsidence over this region. Therefore tape recorder signal in HCN seen in figure-3 may have the source from another region where rising branch of the Walker circulation is located. It will be good to show a vertical variation of HCN in the lower stratosphere averaged over the region of Maritime Continent. It may show a clear picture of tape-recorder signal in HCN during the years other than El-Nino years. Then use this plot for inter-comparison within satellite measurements.

Specific comments: Authors missed adding information of the Walker circulation in the introduction section. Few details on vertical transport into the upper troposphere and lower stratosphere by the convection would be useful. The abstract does not provide complete information about the manuscript.

Section 2.2: Surface measurements at the Jungfraujoch in Switzerland (46.5âŮęN), Mauna Loa in Hawaii (19.3âŮę 15 N), and Kitt Peak, Arizona (32âŮęN), show higher than normal values during El-Nino. This can be justified by transport from Indonesia. However, vertical transport to the stratosphere from Indonesia may not occur during El Nino due to subsidence over this region.

Page 9, line no 8-13. During the normal years (Not El Nino years) MLS show transport from Asian monsoon region if you take an average over the region 20-40N, 60-120E). This is related to vertical transport by monsoon convection as stated by Randel et al 2010, Ploeger et al., 2017). While during an El Nino year, HCN may not get transported to the stratosphere from this region due to subsidence over this region (see Clim Dny DOI 10.1007/s00382-016-3451-6).

Section 3.4 is very poor. I suggest not drawing conclusions from zonal or meridional average plots. Authors should select one specific region and provide details (e.g., Maritime Continent).

I suggest providing (may be from Reanalysis data) a figure of anomalies (year Elnino-Climatology) of vertical winds and circulations over the selected region. It will be helpful

in depicting vertical transport into the lower stratosphere.

Technical corrections (1) Page 1 line no 2: greater than other Elniño events or Climatology? (2) It is difficult to read figure 4 due odd y-axis scale

---

## Author Comment (AC2) · 16 Aug 2017

**1    Reply to general comments**

The reviewer notes that the manuscript is very short and lacks details. This is a matter of opinion and taste.  The paper, if it were published as it is, would run to 13 journal pages; this does not seem short to us.  Furthermore, we have attempted to show the MLS data with as much other information as the reader needs to interpret it, but no more.  We have deliberately kept the material on ENSO to the minimum necessary; there is a huge amount of literature on the subject already.

The reviewer states that the zonal mean plots in figure 3 provide poor information be-

cause they do not show at which longitude the HCN is entering the stratosphere. It is for exactly this reason that we included figure 6, which makes it clear at which longitude the excess HCN first appears in the MLS data. Plots along the lines of figure 3 in the paper look much the same if you make them for a restricted range of longitudes; we show an example in figure 1 of this reply. Note that the two panels of this figure look almost identical except at 100 hPa and 68 hPa; here, the 95°E to 155°E panel shows clearly the sudden arrival of HCN in late autumn of 2015 (and to a lesser extent in 2006 and 2014). In the paper we make this point using figure 6; we did not feel that it was necessary to also include time-height plots for specific longitude ranges. It is unsurprising that tracer mixing ratios in the tropical stratosphere are zonally symmetric except for short periods immediately after large amounts of a tracer cross the tropopause at a specific longitude. The zonal winds in this region are far faster than the meridional winds or the vertical transport, ensuring that any difference between one longitude and another is smoothed out on a timescale of a month or two.

The reviewer makes a valid point in that it might seem surprising that air is entering the stratosphere over the maritime continent, at a time when the Walker circulation has moved so that air is, in general, descending in that region. Nevertheless, the maps in figure 6 of the paper make it clear that the excess HCN appears in the equatorial lower stratosphere at a very restricted range of longitudes around 90°E or 100°E, and does so in a very short period of time.

There is essentially no doubt that the source of the HCN is fires in Indonesia. A trawl through newspaper articles from the time makes it clear that the fires were a serious event, causing widespread regional pollution in the lower troposphere (See https://www.theguardian.com/world/2015/oct/26/indonesias-fires-crime-against-humanity-hundreds-of-thousands-suffer for a typical article.) A second article (https://www.theguardian.com/environment/ng-interactive/2015/dec/01/indonesia-forest-fires-how-the-years-worst-environmental-disaster-unfolded-interactive)

contains some useful animations of the fire season, demonstrating that it ran from early July through to late November, but that the largest emission of CO to the free troposphere occurred in mid to late October. For more detail on the CO, in a refereed publication, see Field et al. (2016), which makes it clear that biomass-burning products were present throughout the troposphere over Indonesia in the latter half of 2015, and that they reached the tropopause in late October. As we explain in the paper, because the burnt material in Indonesia tends to contain large amounts of peat, the ratio of HCN to CO is likely to be larger than in biomass-burning events elsewhere in the world.

In an attempt to assess whether it is reasonable to expect transport from the ground to the stratosphere at the time we observe the excess HCN, we plot in figure 2 of this reply the vertical velocities averaged over the region in which we observed the HCN entering the stratosphere. The data are taken from the NCEP Operational Model Global Tropospheric Analyses (DOI: 10.5065/D6M043C6) and are smoothed with a 15-day binomial smoother. The vertical dotted lines mark 2015-10-24 and 2015-10-31, corresponding to the upper right-hand panel of figure 6 in the discussion paper. It is clear that during June-October 2015 the vertical velocity was mostly downwards, whereas the average over the 2010-2016 period was upwards. However, in all years, the air begins to ascend strongly towards the end of October, at the end of the dry season and the start of the wet season. In 2015 this ascent begins later than average and is not as strong as on average, but it is ascent nevertheless. Furthermore, the change to an overall average ascent (which may have been preceded by more localised ascent episodes) begins at the time that we see the HCN appearing at the tropopause. The period corresponding to the top-right panel of figure 6 in the discussion paper is marked on figure 2 of this reply with two dotted vertical lines. We therefore do not accept the referee's argument that upwards transport is not possible in this region at this time. The actual transport probably occurs in individual convection events, possibly initiated by the fires themselves, but figure 2 of this reply suggests that the average vertical velocity in the region is not such that it would suppress the convection.

[Figure]

**2  Reply to specific comments**

- *Authors missed adding information of the Walker circulation in the introduction section.* We will add a sentence or two, but we do not consider it appropriate to add a detailed description.

- *[A] few details on vertical transport into the upper troposphere and lower stratosphere by the convection would be useful.* It is far from straightforward to provide "a few details" on this in a useful way. In 1981, Newell and Gould-Stewart (1981) suggested that air must enter the stratosphere over the maritime continent in November-March and over the Bay of Bengal during the summer monsoon period. That paper has been cited over 250 times in the intervening years, not always by people who think it is correct (Dessler, 1998). The exact details of how air enters the stratosphere from the troposphere and the extent to which convection plays a role in the process is still an area of ongoing research. A good recent paper on rapid transport paths from the surface in the tropics to the upper troposphere and stratosphere is Hosking et al. (2012) — we will cite this, and add a sentence or two on the subject to the final version.

- *The abstract does not provide complete information about the manuscript.* Indeed, that is the point of an abstract. It is clear that the reviewer feels that there is a part of the paper which is not adequately represented in the abstract, but he does not state which part that is. We therefore intend to leave the abstract unchanged.

- *Section 2.2: Surface measurements at the Jungfraujoch in Switzerland (46.5° N), Mauna Loa in Hawaii (19.3° N), and Kitt Peak, Arizona (32° N), show higher than normal values during El-Nino. This can be justified by transport from Indonesia. However, vertical transport to the stratosphere from Indonesia may not occur during El Nino due to subsidence over this region.* In stating that the Jungfraujoch

data are less easy to explain, we meant that Rinsland et al. (2000) were of that opinion; this is why they present several different explanations. We will re-word that sentence to make it clearer that this is the case. We have already shown why we do not agree with the reviewer's opinion that HCN is unlikely to be transported into the stratosphere over Indonesia during an El-Niño year.

- *Page 9, line no 8-13. During the normal years (Not El Nino years) MLS show transport from Asian monsoon region if you take an average over the region 20-40N, 60-120E). This is related to vertical transport by monsoon convection as stated by Randel et al. 2010, Ploeger et al., 2017). While during an El Nino year, HCN may not get transported to the stratosphere from this region due to subsidence over this region (see [this paper in climate dynamics: http://dx.doi.org/10.1007/s00382-016-3451-6 ]).* It may indeed be possible to see the monsoon-related transport as described by Randel et al. (2010) in the MLS HCN data, but we feel that to extend the current paper to examine that subject would make it too long and too unfocused. As noted above, there is, in fact, ascent and not subsidence at the time we observe HCN transported rapidly into the stratosphere.

- *Section 3.4 is very poor. I suggest not drawing conclusions from zonal or meridional average plots. Authors should select one specific region and provide details (e.g., Maritime Continent).* We are puzzled by this comment. Section 3.4 is the one place in the paper where we show maps, so that the geographical location of a phenomenon can be seen clearly, and where we are not drawing a conclusion from a zonal mean. For most of the time, HCN in the tropical stratosphere is close to zonally symmetric (as in the top left panel of figure 6) and a zonal mean tells you almost everything there is to know about it. The remaining panels of figure 6 show a period which is very unusual in that the HCN distribution is not zonally symmetrical, with much higher values appearing rather suddenly over the maritime continent. Once the excess HCN has arrived in the stratosphere, the

rapid zonal winds disperse it to all longitudes on a timescale of a month or two.

- *I suggest providing (maybe from Reanalysis data) a figure of anomalies (year El Niño-Climatology) of vertical winds and circulations over the selected region. It will be helpful in depicting vertical transport into the lower stratosphere.*

  We show vertical velocities in figure 2 of this reply. We will amend the text of the paper to mention that, although there is descent over the source region from June to October it changes rapidly to ascent at the end of October. We do not feel that it is necessary to include figure 2 of this reply in the final version of the paper.

**3  Reply to technical corrections**

- *Page 1 line no 2: greater than other El-Niño events or Climatology?*  We meant that, in the major El-Niño years, HCN is both higher than the climatological mean and higher than it is in a more moderate El-Niño year such as 2006-7 or 2009-10. Based on the data that we have for major El-Niño years, HCN in such years is higher than at any other time. We will attempt to re-word that sentence so that any ambiguity is reduced.

- *It is difficult to read figure 4 due odd y-axis scale.* The y axis scale has a tick every 20 degrees of latitude, including zero; I am not sure what I should do to improve that. I have used a tick every 50 degrees in figure 5 as the panels are smaller, but I think 20 degrees is better for figure 4. I suspect that the reviewer's problem is the way that R has chosen to put labels on the ticks at -80, -40, 0, +20 and +60, where -80, -40, 0, 40, 80 would seem more symmetrical. We have re-built the figure, setting the labels by hand to be at -80, -40, 0, 40, 80.
**References**

Robert D. Field, Guido R. van der Werf, Thierry Fanin, Eric J. Fetzer, Ryan Fuller, Hiren Jethva, Robert Levy, Nathaniel J. Livesey, Ming Luo, Omar Torres, and Helen M. Worden. Indonesian fire activity and smoke pollution in 2015 show persistent nonlinear sensitivity to El Niño-induced drought. *Proc. Nat. Acad. Sci.*, 113(33):9204–9209, 2016. doi: 10.1073/pnas.1524888113.

Reginald E. Newell and Sharon Gould-Stewart. A stratospheric fountain? *J. Atmos. Sci*, 38: 2789–2796, 1981. doi: 10.1175/1520-0469(1981)038<2789:ASF>2.0.CO;2.

A. E. Dessler. A reexamination of the "stratospheric fountain"hypothesis. *Geophys. Res. Lett.*, 25(22):4165–4168, November 1998. doi: 10.1029/1998GL900120.

J. S. Hosking, M. R. Russo, P. Braesicke, and J. A. Pyle. Tropical convective transport and the Walker circulation. *Atmospheric Chemistry and Physics*, 12(20):9791–9797, 2012. doi: 10.5194/acp-12-9791-2012.

C. P. Rinsland, E. Mahieu, R. Zander, P. Demoulin, J. Forrer, and B. Buchmann. Free tropospheric CO, $C_2H_6$, and HCN above central Europe: Recent measurements from the Jungfraujoch station including the detection of elevated columns during 1998. *J. Geophys. Res.*, 105(D19):24235 – 24249, October 2000. doi: 10.1029/2000JD900371.

William J. Randel, Mijeong Park, Louisa Emmons, Doug Kinnison, Peter Bernath, Kaley A. Walker, Chris Boone, and Hugh Pumphrey. Asian Monsoon Transport of Pollution to the Stratosphere. *Science*, 328(5978):611–613, April 2010. ISSN 0036-8075. doi: 10.1126/science.1182274.
* * *
[Figure]

[Figure]

**Fig. 1.** MLS HCN anomaly (with respect to the 2005-2012 mean) as a function of time and altitude for two different longitude ranges: 95E to 155E and 180W to 95E.

[Figure]

**Fig. 2.** Vertical velocity at 500 hPa averaged over a region between 90E and 120E, and between 6S and 6N. Note that, as the data are in units of Pa/s, negative values mean ascent, positive values descent.

---

## Referee Comment (RC2) · M. Tao (Referee) · 25 Aug 2017

The study highlights a record-breaking MLS HCN increase (for the Aura mission period) in the tropical lower stratosphere during 2015-2016 El-Nino event. Other two satellite datasets (ACE-FTS and MIPAS) as well as the ground based measurements show reasonable agreements with MLS, which supports sufficiently the main conclusion of the paper. The paper presents important results and should be in good shape for publication in ACP. I recommend its publication with more discussion about transport features into stratosphere related to warm ENSO events.

Specific comments:

1. The authors suggest the main driver of enhanced HCN is droughts in equatorial

[Figure]

Asia during El-Nino. However, the connection between warm ENSO and transport in tropical UTLS is not mentioned. I suggest adding one paragraph to review the state-of-art about the association between ENSO and troposphere to stratosphere transport. For example, 1) the convective transport over Maritime Continents; 2) stronger TTL upwelling during El Niño winters (e.g. Randel et al. 2009, Calvo et al.Âă2010, Konopka et al. 2015).

2. MIPAS data shows a second maximum of HCN around 20-30 N on 100 hPa and 68 hPa firstly in boreal winter but not in MLS in Figure 5. The authors did not comment on this feature but only comment on the one in NH during summer. Could the author comment on this maximum from the third observation and from the possible source?

3. Still about the section 3.3, the author mentioned the boreal summer increased HCN in NH possibly links to ASM suggested by Randel et al. [2010] and Ploeger et al. [2017]. But why this boreal summer enhanced HCN show clear increase in 2007 and 2016, which do not experience strong AM circulation?

Technical comments:

Page 3 line 3, 'since 1950' instead of 'between 1950 and now'

Page 13 line 10, 'droughts' instead of 'draughts'

---

## Author Comment (AC3) · 7 Sep 2017

**1   Reply to general comments**

The reviewer is in general happy with the paper, but asks us to add more discussion about transport features into stratosphere related to warm ENSO events. We will endeavour to add a paragraph on this.

**2  Reply to specific comments**

- *The authors suggest the main driver of enhanced HCN is droughts in equatorial Asia during El-Niño. However, the connection between warm ENSO and transport in tropical UTLS is not mentioned. I suggest adding one paragraph to review the state-of- art about the association between ENSO and troposphere to stratosphere transport. For example, 1) the convective transport over Maritime Continents; 2) stronger TTL upwelling during El Niño winters (e.g. Randel et al. 2009, Calvo et al., 2010, Konopka et al. 2015).* This request is similar to that made by reviewer 1; we will endeavour to add a suitable paragraph. We thank the reviewer for the references.

- *MIPAS data shows a second maximum of HCN around 20-30° N on 100 hPa and 68 hPa firstly in boreal winter but not in MLS in Figure 5. The authors did not comment on this feature but only comment on the one in NH during summer. Could the author comment on this maximum from the third observation and from the possible source?* It is not clear to us what causes the feature that is noted by the reviewer. The feature appears in the 2006-7 NH winter, which is an El Niño winter, but the other time is is visible is the 2010-11 winter, at which time the ENSO index is negative. The 2006-7 NH maximum probably has the same source as the SH maximum: El-Niño-related fires in Indonesia. In 2010-11 there was unusually high levels of burning in South America (Glatthor et al., 2015). These are responsible for the SH high values and may be responsible for the NH values as well. We would prefer not to add any material on this to the present paper as the MIPAS data are already described in detail by Glatthor et al. (2015). The main reason for showing the MIPAS data in the present paper is to show the extent to which the MLS data are credible.

- *Still about the section 3.3, the author mentioned the boreal summer increased HCN in NH possibly links to ASM [Asian Summer Monsoon] suggested by Ran-*

*del et al. [2010] and Ploeger et al. [2017]. But why this boreal summer enhanced HCN show clear increase in 2007 and 2016, which do not experience strong AM circulation?* We do not know for certain, but a reasonable hypothesis is as follows. To observe enhanced HCN over the ASM requires a strong AM circulation and/or increased quantities of HCN in the troposphere. 2007 and 2016 may not have had a strong AM circulation, but they certainly had increased quantities of HCN in the troposphere, following the unusual biomass burning in late 2006 and late 2015. Although some of this HCN reaches the stratosphere very rapidly, as we show in this paper, much of it remains in the troposphere. Away from the oceanic boundary layer the lifetime of HCN is long enough for there to be high levels to be lofted by the monsoon circulation, 6-8 months after it was emitted by biomass burning.

**3  Reply to technical corrections**

- *Page 3 line 3, 'since 1950' instead of 'between 1950 and now'.* That is a better way to word the sentence; we will use it.

- *Page 13 line 10, 'droughts' instead of 'draughts'* We will make that correction.

**References**

N. Glatthor, M. Höpfner, G. P. Stiller, T. von Clarmann, B. Funke, S. Lossow, E. Eckert, U. Grabowski, S. Kellmann, A. Linden, K. A. Walker, and A. Wiegele. Seasonal and interannual variations in HCN amounts in the upper troposphere and lower stratosphere observed by MIPAS. *Atmospheric Chemistry and Physics*, 15(2):563–582, 2015. doi: 10.5194/acp-15-563-2015.

---

## Author Response (AR1)

**List of corrections and changes to acp-2017-602**

Hugh Pumphrey and co-authors

October 25, 2017

**1 Replies to referees' comments**

We made detailed replies to the referees' comments in author comments AC2 and AC3; we therefore do not repeat those replies here.

**2 Significant changes to text**

- Added a little on the Walker circulation to section 2.1 as requested by Reviewer 1. A citation to Bjerkes's 1969 paper in which the term "Walker circulation" was coined is included.

- Expanded text at the end of section 4 as described in author comment AC1.

- Paragraph added to section 4 explaining that the ascent happened at the end of the wet season as the circulation switches from descent to ascent over Indonesia.

- Text added to section 4 to add a little on vertical transport and a reference to Hosking et al. (2012).

- Figures 3, 4, 7 and 8 re-made to show the most recent MLS, ACE-FTS, MEI and GFED data.

**3 Technical corrections etc.**

- Slight re-wording of the first sentence of the abstract in response to reviewer 1.

- Re-worded the last sentence of sec 2.2 as requested by reviewer 1.

- Page 3 Line 3: "between 1950 and now" replaced by "since 1950" as requested by reviewer 2.

- Page 13 line 10, "droughts" instead of "draughts" as requested by reviewer 2.

- Latitude axis labels in Figure 4 forced to be at -80, -40, 0, 40, 80 in response to a comment from reviewer 1.

[revised manuscript text omitted]